# A proline switch explains kinetic heterogeneity in a coupled folding and binding reaction

Franziska Zosel [1,3], Davide Mercadante[1], Daniel Nettels[1] & Benjamin Schuler [1,2]

The interactions of intrinsically disordered proteins (IDPs) with their molecular targets are essential for the regulation of many cellular processes. IDPs can perform their functions while disordered, and they may fold to structured conformations on binding. Here we show that the *cis/trans* isomerization of peptidyl—prolyl bonds can have a pronounced effect on the interactions of IDPs. By single-molecule spectroscopy, we identify a conserved proline residue in NCBD (the nuclear-coactivator binding domain of CBP) whose *cis/trans* isomerization in the unbound state modulates the association and dissociation rates with its binding partner, ACTR. As a result, NCBD switches on a time scale of tens of seconds between two populations that differ in their affinities to ACTR by about an order of magnitude. Molecular dynamics simulations indicate as a cause reduced packing of the complex for the *cis* isomer. Peptidyl-prolyl *cis/trans* isomerization may be an important previously unidentified mechanism for regulating IDP interactions.

[1] Department of Biochemistry, University of Zurich, Winterthurerstrasse 190, 8057 Zurich, Switzerland. [2] Department of Physics, University of Zurich, Winterthurerstrasse 190, 8057 Zurich, Switzerland. [3]Present address: Novo Nordisk A/S, Novo Nordisk Park 1, 2760 Måløv, Denmark. Correspondence and requests for materials should be addressed to B.S. (email: schuler@bioc.uzh.ch)

ntrinsically disordered proteins (IDPs) and disordered regions are abundant in the eukaryotic proteome. They are particularly prevalent in regulatory processes such as signal transduction, transcription, and many other complex interaction networks[1–3]. Several explanations have been put forward as to why disorder can be beneficial. For instance, structural flexibility enables the interaction with multiple binding partners[4] and facilitates interactions with both high association and dissociation rates[5], rendering signaling processes more rapid. Moreover, disordered segments are easily accessible for post-translational modifications that modulate interactions and the corresponding signaling pathways[6].

The conformational heterogeneity underlying the properties of IDPs results from structurally diverse ensembles that are sampled on a broad range of timescales[7–9]. The fastest long-range dynamics occur in the sub-microsecond range and correspond to the reconfiguration of the disordered polypeptide chain, sometimes impeded by residual non-native interactions causing internal friction[10]. Transient secondary structure formation typically occurs on the microsecond time scale[11], and more persistent long-range interactions in IDPs can lead to dynamics in the millisecond range[12]. All of these dynamics are expected to be faster than the typical kinetics of binding at cellular protein concentrations[13,14]. As a result, it is usually reasonable to assume a separation of timescales between the internal dynamics of an IDP and its association and dissociation kinetics. In this case, the binding kinetics can be approximated by a two-state process with single-exponential relaxation, as frequently observed experimentally[14–16].

There is, however, a source of slow conformational dynamics in IDPs whose influence on binding kinetics has hardly been investigated, but that is known to occur on timescales of seconds to minutes: the *cis/trans* isomerization of peptide bonds involving proline residues[17,18]. Proline isomerization is a classic source of slow phases in protein folding kinetics[19], and given that Pro is the amino acid most highly enriched in IDPs compared to folded proteins[20], it may be expected to influence the behavior of many IDPs and cause heterogeneity in their association and dissociation kinetics. Here we use single-molecule Förster resonance energy transfer (FRET) to probe the kinetic heterogeneity and its structural origin for the interaction between the nuclear-coactivator binding domain (NCBD) of the CBP/p300 transcription coactivator and the activation domain of SRC-3 (ACTR). NCBD and ACTR are a classic example of two IDPs that undergo a coupled folding and binding reaction in which both proteins gain structure and form a cooperatively folded core in the complex[4,21]. Unbound ACTR contains only little residual secondary structure[22]. In contrast, unbound NCBD is a marginally stable, molten-globule-like protein, with a large content of α-helical structure[23,24], and different arrangements of the helices have been observed in complexes with its different binding partners[21,25–27]. We find that NCBD occupies two distinct subpopulations at equilibrium that differ by the conformation of one proline residue. Both the *trans-* and the *cis*-Pro populations are able to bind ACTR, but with different affinities and kinetics, thus revealing a potentially widespread source of slow dynamics and heterogeneity in IDP interactions that may have an important role in regulation.

## Results

### Single-molecule FRET of immobilized molecules.
Investigating heterogeneous kinetics that span a broad range of timescales with single-molecule spectroscopy requires individual molecules to be observed for extended periods of time with high time resolution. We thus immobilized biotinylated NCBD labeled with Cy3B as a donor fluorophore on a polyethylene glycol- (PEG-) passivated glass surface via avidin (see Methods) and recorded fluorescence emission from single molecules by confocal single-photon counting. Including low concentrations of ACTR (labeled with CF680R as an acceptor fluorophore) free in solution allowed us to monitor association and dissociation events via FRET in great detail (Fig. 1a). In the absence of bound ACTR, only donor emission is observed from a single immobilized NCBD molecule (plus a small background contribution from direct excitation of labeled ACTR molecules in solution). Upon binding of an ACTR molecule, FRET results in a decrease in donor fluorescence and an increase in acceptor fluorescence (Fig. 1b). Upon dissociation of ACTR, FRET ceases and acceptor emission is lost. At a typical ACTR concentration, $c_{ACTR}$, of 65 nM used here (determined from a fluorescence correlation spectroscopy (FCS) measurement in the solution above the surface; see Methods), several association and dissociation events revealed by such anticorrelated changes in donor and acceptor signal occur each second (Fig. 1). With an average duration of the single-molecule time traces of ~1 min until the donor dye photobleaches (Supplementary Fig. 2e), we can thus record hundreds to thousands of binding and dissociation events of a single NCBD molecule, yielding excellent statistics even from individual time traces (Fig. 1, Supplementary Fig. 1, Supplementary Fig. 2). Note that acceptor photobleaching is not an issue, since the dissociation of ACTR is much faster than photobleaching (<0.1 s$^{-1}$), and new ACTR molecules are always replenished from the solution.

### Slowly interconverting states with different ACTR affinity.
With this approach, the interaction kinetics of ACTR and NCBD can be analyzed at equilibrium from milliseconds to minutes by extracting association and dissociation rate coefficients from the distributions of dwell times in the bound and unbound states or by using maximum likelihood (MLH) analysis in combination with hidden Markov models[28,29], as we will show below. However, besides the rapid transitions between the bound and the unbound state, inspection of the time traces reveals two kinetic regimes that alternate on a much slower time scale: in the time trace of Fig. 1b, e.g., the binding events between ~5 s and ~59 s last much longer on average than outside that period. Correspondingly, the relative population of the bound state is increased during this time (Fig. 1c). This behavior is observed in many time traces (Supplementary Fig. 1c) and reveals that a single NCBD molecule can switch between two kinetic regimes that persist for tens of seconds. Notably, the transfer efficiencies are very similar in both kinetic regimes (Fig. 1c), so only their kinetic characteristics—and not the photon count ratios or transfer efficiencies—can be used to distinguish them.

To illustrate the difference in binding kinetics between the two regimes quantitatively, we split all time traces into 4-second segments and determined for each segment the mean dwell time of NCBD in the bound state, $\langle\tau_{on}\rangle$, and in the unbound state, $\langle\tau_{off}\rangle$, using the Viterbi algorithm (see Methods). The corresponding 2D histogram of dwell times (Fig. 1d) shows two populations that reflect the two kinetic regimes. We used the resulting average dwell times as initial values for a global MLH analysis of all recorded time traces with a hidden Markov model based on the kinetic scheme in Fig. 1e. The model describes association and dissociation of ACTR and NCBD, with NCBD additionally interconverting between two states, NCBD1 and NCBD2, that bind ACTR with different kinetics. The resulting rate coefficients (Table 1) translate into equilibrium dissociation constants ($K_{d,1} = k_{off,1}/k_{on,1}$ and $K_{d,2} = k_{off,2}/k_{on,2}$) of ACTR to NCBD1 and NCBD2, respectively, that differ by about an order of magnitude, with $78 \pm 6$ nM for the high-affinity state, NCBD1,

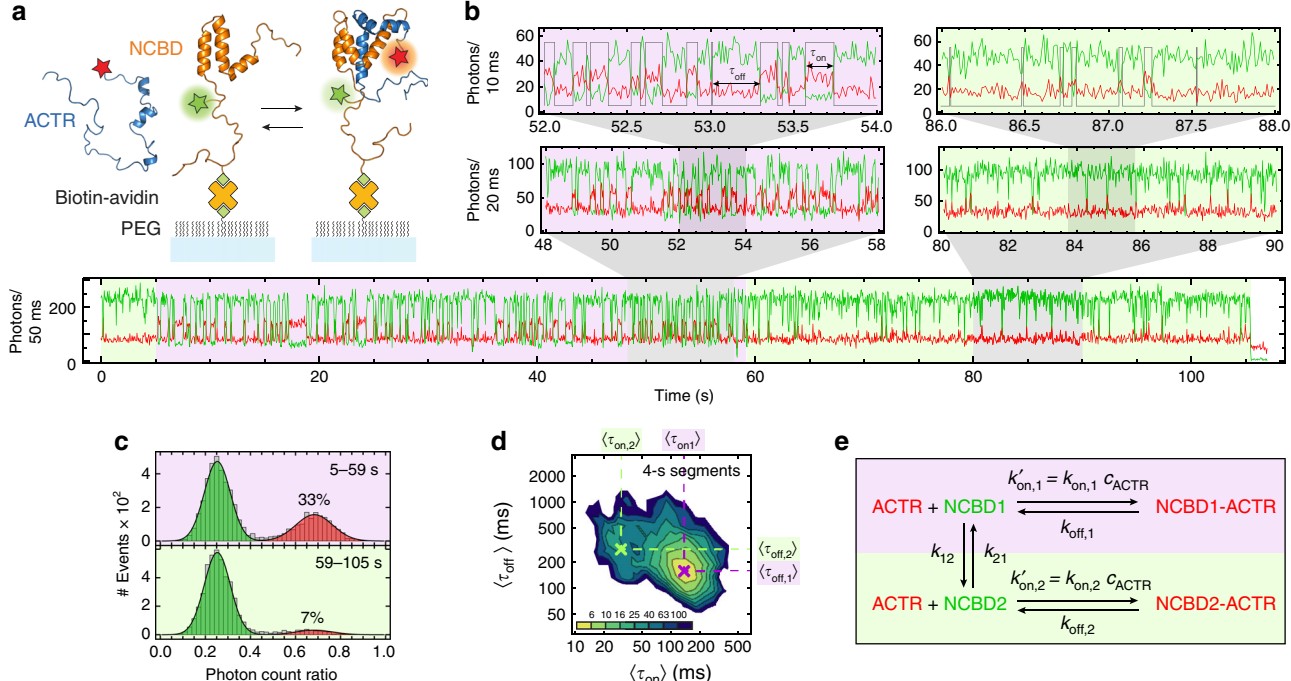

**Fig. 1** Binding of ACTR to single surface-immobilized NCBD molecules. **a** Schematic representation of acceptor-labeled ACTR (blue) binding to surface-immobilized donor-labeled NCBD (orange). **b** A single-molecule time trace showing donor (green) and acceptor fluorescence (red) until donor photobleaching, with progressive magnifications above. A period where ACTR forms relatively long-lived complexes with NCBD is shaded in purple; binding events are sparser and on average shorter in the segments shaded in light green. The most likely state trajectory (bound/unbound) identified by the Viterbi algorithm is depicted as gray lines for the largest magnification. 283 binding events were identified in this time trace. Note that the lower quantum yield of the acceptor compared to the donor dye leads to a decrease in total fluorescence intensity upon ACTR binding. **c** Histograms of photon count ratios (acceptor over total number of photons) for the two kinetic regimes in **b**, calculated for 20-ms time bins. The two populations arise from the unbound (green, low photon count ratio) and the bound state (red, high photon count ratio) of NCBD; the corresponding fractions bound and the time ranges in the trajectory are indicated. **d** 2D histogram of the mean time in the bound state, $\langle \tau_{on} \rangle$, and in the unbound state, $\langle \tau_{off} \rangle$, generated by splitting all 163 measured time traces into 4-s segments, resulting in two clusters. **e** Kinetic binding model used to analyze the time traces, including the interconversion between the two states of NCBD with different kinetics of dissociation and association with ACTR

**Table 1 Kinetic parameters determined with maximum likelihood analysis of single-molecule FRET experiments**

| Immobilized protein | $k'_{on,1} = k_{on,1} c_{ligand}$ (s$^{-1}$)[a] | $k'_{on,2} = k_{on,2} \cdot c_{ligand}$ (s$^{-1}$)[a] | $k_{off,1}$ (s$^{-1}$) | $k_{off,2}$ (s$^{-1}$) | $c_{ligand}$ (nM)[b] | $k_{on,1}$ (10$^8$ M$^{-1}$s$^{-1}$)[c] | $k_{on,2}$ (10$^8$ M$^{-1}$s$^{-1}$)[c] | $k_{12}$ (s$^{-1}$) | $k_{21}$ (s$^{-1}$) |
|---|---|---|---|---|---|---|---|---|---|
| NCBD | 6.0 ± 0.2 | 3.0 ± 0.5 | 7.3 ± 0.3 | 30 ± 3 | 65 | 0.93 ± 0.06 | 0.46 ± 0.08 | 0.04 ± 0.01 | 0.07 ± 0.01 |
| ACTR | 3.3 ± 0.3 | 1.6 ± 0.3 | 5.6 ± 0.5 | 30 ± 6 | 17 | 3.2 ± 0.5 | 2.5 ± 0.6 | n.a. | n.a. |
| NCBD P20A | 5.1 ± 0.2 | n.a. | 23 ± 1 | n.a. | 65 | 0.78 ± 0.05 | n.a. | n.a. | n.a. |

Errors are the standard deviations of ten bootstrapping trials if not stated otherwise
n.a., not applicable
[a]Pseudo-first-order association rate coefficient observed in the time traces. The second-order association rate coefficient is calculated based on the ligand concentration $c_{ligand}$ measured by FCS in solution (see Methods)
[b]Concentrations of the acceptor-labeled ligand were determined using FCS (see Methods). An uncertainty of ±5% was estimated from two independent measurements conducted before and after recording the time traces
[c]Second-order association rate coefficient calculated from the pseudo-first-order association rate coefficient based on the ligand concentration, $c_{ligand}$, measured by FCS in solution (see Methods). Uncertainty from propagating the error of $k'_{on}$ and the ligand concentration. The error for immobilized ACTR is greater owing to the greater uncertainty in the relative populations of NCBD1 and NCBD2, $p_1$ and $p_2$: $p_1 = k_{21}/(k_{12} + k_{21})$ and $p_2 = k_{12}/(k_{12} + k_{21})$

and 650 ± 150 nM for the low-affinity state, NCBD2 (errors are standard deviations based on bootstrapping). The interconversion rates between NCBD1 and NCBD2 are slow ($k_{12} = 0.04 ± 0.01$ s$^{-1}$ and $k_{21} = 0.07 ± 0.01$ s$^{-1}$) (Table 1), as expected. If direct interconversion between the bound states, NCBD1-ACTR and NCBD2-ACTR, is included in the model, MLH analysis returns values for the corresponding rate coefficients indistinguishable from zero; we thus exclude this link.

To test the proposed kinetic model further, we take advantage of the possibility to invert the roles of the two binding partners in the single-molecule experiments and probe association of surface-immobilized ACTR with freely diffusing NCBD (Fig. 2). This

arrangement allows us to assess whether ACTR exhibits similar kinetic heterogeneity as NCBD, with long-lived interconverting states. Such behavior is absent in the 75 time traces recorded (see Fig. 2b and Supplementary Fig. 3a for examples). Consistently, a 2D dwell-time histogram for time trace segments of 4 s shows only one population (Fig. 2d), indicating the presence of a single, kinetically homogenous population of ACTR. However, if NCBD exists in a high- and a low-affinity state as proposed above, the reaction should be described by the kinetic model in Fig. 2e, where two different NCBD populations bind to ACTR. The model predicts single-exponential association kinetics decaying with the sum of the two pseudo-first-order

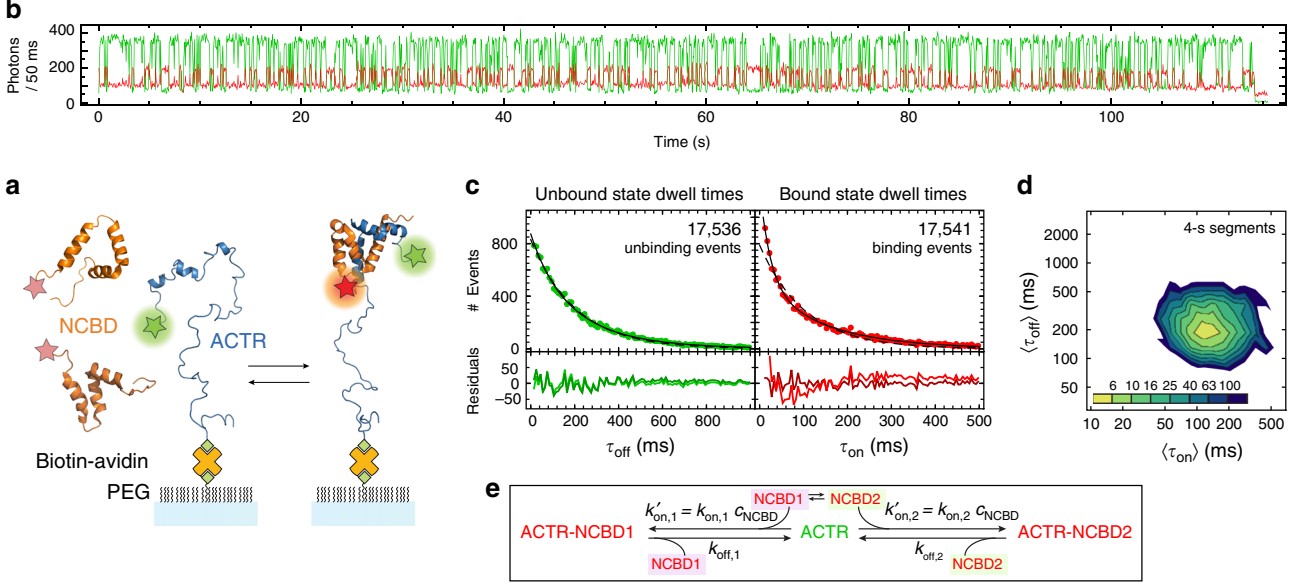

**Fig. 2** Binding of NCBD to single surface-immobilized ACTR molecules. **a** Schematic representation of acceptor-labeled NCBD (orange) binding to surface-immobilized donor-labeled ACTR (blue). **b** Representative single-molecule time trace (donor emission green, acceptor emission red). **c** Dwell-time distributions of the unbound state (green) and the bound state (red) with single-exponential (dashed lines, light residuals) and double-exponential fits (solid lines, dark residuals). **d** 2D histogram of $\langle \tau_{on} \rangle$ and $\langle \tau_{off} \rangle$ from 75 time traces, showing a single cluster. **e** Kinetic model used to analyze the time traces, with two different NCBD states interacting with ACTR

association rate coefficients, $k'_{on,1} + k'_{on,2}$ ($k'_{on,1} = k_{on,1} \cdot c_{NCBD1}$ and $k'_{on,2} = k_{on,2} \cdot c_{NCBD2}$, where $k_{on,1/2}$ are the second-order rate coefficients and $c_{NCBD1/2}$ the concentrations of NCBD1/2 free in solution). The dissociation kinetics, however, are predicted to be bi-exponential, with the two phases (with $k_{off,1}$ and $k_{off,2}$) arising from the high- and low-affinity NCBD-ACTR complexes, respectively[30]. Indeed, this is what we observe (Fig. 2c), and a global MLH analysis of all time traces reveals dissociation rate coefficients in good agreement with those observed for immobilized NCBD (Table 1). Moreover, the observed fraction of fast transitions [given by $k_{12}k_{on,2}/(k_{21}k_{on,1} + k_{12}k_{on,2})$] for immobilized ACTR (32 ± 4%) is close to the corresponding value from the measurements with immobilized NCBD (24 ± 6%). This agreement supports the presence of the two different states of NCBD and furthermore suggests that surface interactions do not have a pronounced effect on the measurements.

We note, however, that the association rate coefficients, $k_{on,1}$ and $k_{on,2}$, are three to five times greater when ACTR is immobilized rather than NCBD (cf. Table 1), in better agreement with published values[31]. To test whether this discrepancy is caused by NCBD immobilization, we performed stopped-flow experiments with the same constructs used for immobilization (Supplementary Fig. 4). The resulting association rate coefficients [(0.89 ± 0.07)·10^8 M^−1 s^−1 for ACTR binding to biotinylated NCBD and (3.0 ± 0.1)·10^8 M^−1 s^−1 for NCBD binding to biotinylated ACTR; an average from the contributions of the high- and the low-affinity population] agree well with the ones obtained from the single-molecule experiments, indicating that the dominant effect on the kinetics originates from biotinylation and not surface immobilization per se. A likely cause is the negative net charge (−3) of the Avi-tag introduced for biotinylation in vivo, which either reduces (in the case of NCBD-Avi) or increases (in the case of ACTR-Avi) the charge-promoted interaction between ACTR and NCBD (net charges of −8 and +6, respectively)[31]. The charges of the dyes (Cy3B: 0, CF680R: −1) appear to have a smaller influence than the Avi-Tag.

**Pro20 isomerization induces a conformational switch in NCBD.** What is the molecular origin of the long-lived kinetic heterogeneity in NCBD? Given the marginal stability of NCBD, the presence of conformational states that persist for seconds may be surprising. However, NCBD contains seven proline residues, four of which are located in the central part of the protein, where they terminate helices (Fig. 3a). Moreover, the time scale of peptidyl-prolyl *cis/trans* isomerization (~0.06 s^−1 at 22 °C in unfolded peptides[32,33]) is in accord with the slow dynamics observed in NCBD. A first corroboration of the hypothesis that peptidyl-prolyl *cis/trans* isomerization causes the slow kinetic heterogeneity comes from the addition of the peptidyl-prolyl *cis/trans* isomerase Cyclophilin A. Cyclophilin A accelerates the switching between low- and high-affinity subpopulations of NCBD ($k_{12}$ and $k_{21}$) by almost an order of magnitude, while leaving the association rate coefficients unaffected (Supplementary Fig. 5, Supplementary Table 1). But which proline residue(s) cause(s) the observed switching?

To address this question, we prepared NCBD variants with Pro to Ala substitutions and tested their interaction in single-molecule FRET experiments with immobilized ACTR, analogous to Fig. 2. The dissociation kinetics of the nine NCBD variants with different combinations of proline substitutions fall into two classes: all variants where Pro20 is substituted by alanine (P20A) decay single-exponentially, whereas the variants containing P20 decay bi-exponentially (Fig. 3b, c; for rate coefficients, see Supplementary Table 2). Pro20 is thus the most likely cause of the slow kinetics observed in NCBD. To test the importance of Pro20 also in the inverse experiment, we incorporated the P20A substitution into surface-immobilized NCBD and followed the binding kinetics of freely diffusing ACTR, analogous to Fig. 1. As expected, both binding and dissociation kinetics are single-exponential and can be described by a simple two-state model (Fig. 3d, f, h), with dissociation rate coefficients similar to measurements where ACTR is immobilized and NCBD lacking Pro20 is free in solution. Most importantly, the time traces with immobilized NCBD P20A no longer exhibit the slow switching

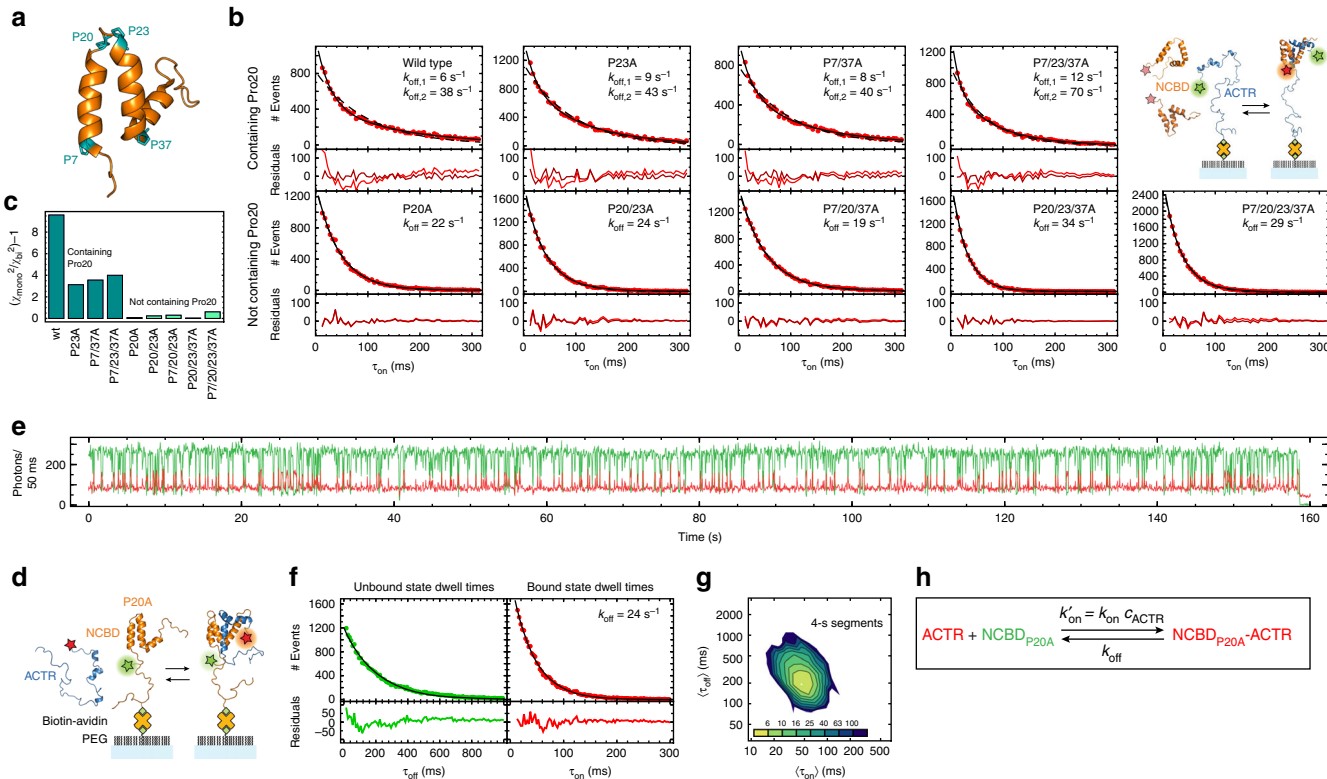

**Fig. 3** Isomerization of Pro20 is the cause of kinetic heterogeneity. **a** Model of NCBD (PDB entry 2KKJ)[23], with helix-terminating proline residues highlighted in turquoise. **b** Dwell-time distributions for the bound state of surface-immobilized ACTR for all NCBD variants, with single-exponential (dashed line, light red residuals) and double-exponential fits (solid line, dark red residuals) and the fitted rates indicated. Dwell times below 12 ms are too short to be identified reliably by the Viterbi algorithm (Supplementary Fig. 6) and are hence omitted. MLH analysis results are compiled in Supplementary Table 2. **c** Normalized ratio of chi-squared as an indicator for the quality of single- and double-exponential fits in **b**, showing variants containing Pro20 in dark green and those with Pro20 exchanged by Ala in light green. **d** Schematic representation of acceptor-labeled ACTR (blue) binding to surface-immobilized donor-labeled NCBD P20A (orange). **e** Corresponding representative single-molecule time trace (donor emission green, acceptor emission red). **f** Dwell-time distributions and residuals for ACTR binding to and dissociating from surface-immobilized NCBD P20A, with single-exponential fits (solid lines). **g** 2D histogram of $\langle \tau_{on} \rangle$ and $\langle \tau_{off} \rangle$ from 113 time traces showing a single kinetic cluster. **h** Two-state kinetic model used to analyze the time traces of NCBD P20A

between the two kinetic regimes (Supplementary Fig. 3b), and splitting the time traces into 4-s segments reveals only a single kinetic cluster (Fig. 3g). In summary, we thus conclude that *cis/trans* isomerization of Pro20 in NCBD causes the protein to switch between two conformational ensembles with different affinities for ACTR.

**Loss of complex stability in the *cis*-state.** To probe the structural origin of the experimentally observed differences in the affinity of the NCBD-ACTR complex depending on the isomerization state of Pro20, we employed molecular dynamics (MD) simulations to sample the dynamics of the complex in the *trans* and the *cis* state of Pro20. Modeling the *cis* state of Pro20 by changing the value of the S19-P20 ω-angle from 180° to 0° would disrupt the topology of NCBD in the complex. Since the inter-dye distances in the complexes of the *trans* and *cis* forms are very similar (Fig. 1c), the arrangement of the NCBD helices in complex with ACTR is unlikely to be altered. We thus modeled the *cis* state by perturbing the ω-angle by well-tempered metadynamics (see Methods for details), which shows an energy barrier between the two states of ~80 kJ mol⁻¹, in line with experimental results[34] and previous calculations using the AMBER2003 force field[35]. We then simulated the *cis* and *trans* conformations separately by replica exchange MD to investigate the stability of the complex when P20 is either in the *trans* or the *cis* state. Within the simulated time and at the chosen temperatures, the simulations do not sample prolyl isomerization, as expected[36] (Supplementary Fig. 7). The

simulations suggest that both ACTR and NCBD are consistently more dynamic when Pro20 is in the *cis* state (Fig. 4a), as indicated by greater root mean square fluctuations in *cis* along the entire sequence of both NCBD and ACTR (Fig. 4b). As a result, the distribution of the radius of gyration of the complex is shifted to slightly higher values with Pro20 in *cis* (Fig. 4c).

This increase in fluctuations and dimensions results from a loss of interactions between the two partners: both the number of hydrogen bonds and of all interactions between atomic pairs within 0.35 nm are on average greater in the *trans* state (Fig. 4d, e). A more detailed analysis indicates a loss of contacts in the loop of NCBD containing Pro20, which is accompanied by a reduced average contact time in the interface between the C-termini of the proteins, as well as a slight loss of packing in the hydrophobic core (Supplementary Fig. 8). Altogether, these findings indicate that a change in the isomerization state of Pro20 from *trans* to *cis* can generate long-range effects that perturb interactions in the NCBD-ACTR complex and lead to faster dissociation and reduced affinity.

## Discussion
We find that NCBD undergoes slow conformational switching between two subpopulations that differ by the conformation of a proline residue. *Cis/trans*-isomerization as the origin of the kinetic heterogeneity observed in single-molecule experiments is confirmed by proline substitutions and the effect of proline *cis/trans*-isomerase on the kinetics. The overall kinetic mechanism (Fig. 5) thus involves

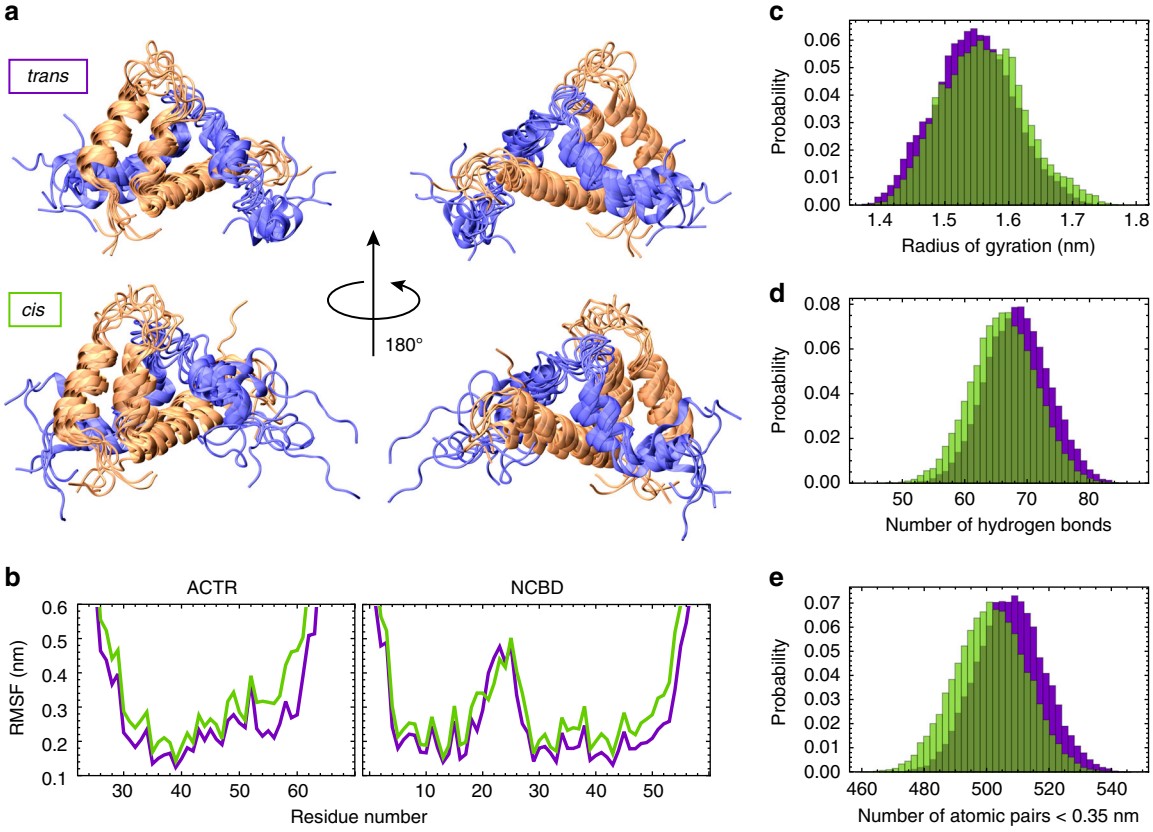

**Fig. 4** Effect of Pro20 isomerization from replica exchange MD simulations. **a** Overlap of representative conformations of the NCBD (orange)-ACTR (blue) complex with Pro20 in the *trans* and *cis* state. **b** Root mean square fluctuation (RMSF) profiles along the sequence of ACTR and NCBD, with NCBD Pro20 in *trans* (purple) or *cis* (green). The RMSF was averaged over all atoms of each residue. **c** Probability distributions for the radii of gyration of the NCBD-ACTR complex, with NCBD Pro20 in *trans* (purple) or *cis* (green). **d**, **e** Probability distributions of intermolecular hydrogen bonds (**d**) or any intermolecular atomic pair falling within a 0.35-nm distance cutoff (**e**) in the MD simulations with Pro20 in *trans* (purple) or *cis* (green)

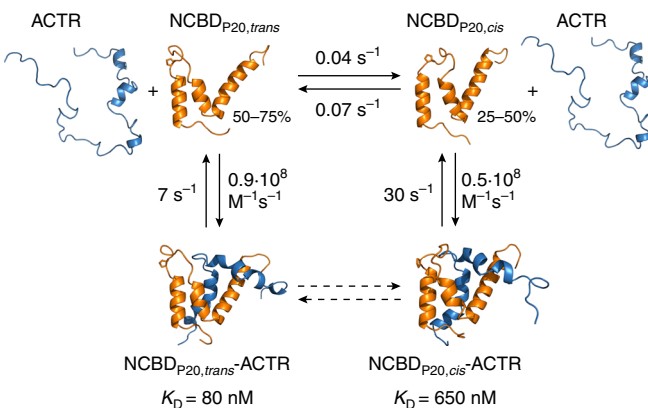

**Fig. 5** Kinetic model of the NCBD-ACTR interaction. The two subpopulations of NCBD differ by the *cis/trans* isomers of the peptide bond involving Pro20, which are interconverting slowly; both bind ACTR, but with different association rate coefficients. The two complexes can be distinguished experimentally by their dissociation rate coefficient, which is greater for the NCBD$_{P20,cis}$-ACTR complex. The difference in association and dissociation rates translates into an eightfold higher affinity of the NCBD$_{P20,trans}$-ACTR complex

four states: NCBD with Pro20 in *trans*, which binds ACTR with higher affinity, and NBCD with Pro20 in *cis*, which binds ACTR with lower affinity. This difference in affinity is largely caused by a change in dissociation rate coefficient; the association rate

coefficients are similar for both subpopulations. The binding interface between ACTR and NCBD has been shown to be sub-optimal and frustrated[37], which might explain why the affinity loss is not even larger in response to the proline switch. We find no evidence for conformational exchange between the *cis*- and *trans*-isomers of Pro20 while NCBD is bound to ACTR. It seems likely that binding locks NCBD in the respective conformation by raising the energy barrier for *cis/trans* isomerization, but we cannot exclude a low rate of exchange that is beyond our current detection limit. Based on the parameters we obtained here (Fig. 5), the *cis* complex is only populated to 6% at equilibrium at the concentrations of 3 mM NCBD and 6 mM ACTR where the NMR structure was solved - consistent with the absence of a population with Pro20 in *cis* in the structure[21]. In unbound NCBD, however, evidence for the presence of *cis/trans* isomerization of Pro20 has been reported based on peak splitting of NMR resonances[24,38]. Peak splitting is also observed in the spectra reported by Kjaergaard et al.[23] if analyzed at lower contour levels (K. Teilum and M. Kjaergaard, personal communication). Moreover, multi-exponential kinetics have been observed in the association and dissociation of the NCBD-ACTR complex with stopped-flow techniques[39], suggesting the presence of a complex interaction mechanism and/or multiple states of the interacting molecules[40], in accordance with our findings.

The interaction between NCBD and ACTR is a prime example for intrinsic disorder of both ligands that enables the formation of transient high-affinity interactions by combining very fast, charge-promoted association rates with dissociation on the milliseconds to seconds time scale[2,41]. NCBD has more than ten

known interaction partners[24,42], among them the activation domains of the steroid receptor coactivator (SRC) family[21], the p53 activation domain[43], and viral factors that hijack the host transcription machinery (e.g., adenoviral E1A)[44]. Rapid dissociation facilitates a fast response of the coactivator machinery to this wide variety of regulators. Structures of NCBD in complex with its other interaction partners reveal large variations, with very different arrangements of the helices, e.g., in the IRF-3[26] and ACTR-bound states[21]. Peptidyl-prolyl isomerization raises the interesting possibility of modulating the interaction with different ligands and thus differentially regulating downstream signaling. Similar to ACTR, other disordered targets of NCBD may bind both subpopulations, but some might have higher affinity for the *cis* conformation. Structured ligands, such as the pointed domain of Ets-2[42] or IRF-3, might bind to one conformation only, as they cannot as readily adapt to different binding interfaces. The high evolutionary conservation of Pro20 in NCBD (Supplementary Fig. 9) supports such an important functional role. Post-translational modifications of the phosphorylation sites reported for NCBD[45] could further regulate the equilibrium and dynamics between the subpopulations and subsequent ligand binding.

Several cases where peptidyl-prolyl *cis/trans* isomerization regulates binding have been described in the context of IDPs. One example is the C-terminal disordered tail of RNA-Polymerase II, where *cis/trans* isomerization of Pro6 (regulated by phosphorylation of the preceding serine residue) allows binding to different downstream factors[46]. Other examples include p53, where a *cis* proline mediates the apoptosis-triggering interaction with BAX[47], as well as the disordered transactivation domain of BMAL-1, where the switch is linked to the circadian rhythm[48]. Furthermore, in an antibody-IDP interaction, proline *cis/trans* isomerization was linked to slow kinetic phases in binding[49]. Considering the high relative abundance of proline residues in IDPs[20], this mechanism of regulating differential binding could be widespread. Elucidating the underlying kinetic mechanisms will be essential for understanding how such regulation by peptidyl-prolyl *cis/trans* isomerization occurs and how it couples to other contributions, such as secondary structure content[50]. Single-molecule spectroscopy, which has previously been employed to expose molecule-to-molecule variation in the dynamics of nucleic acids[51,52] and the influence of proline isomerization on protein folding dynamics[53], provides a versatile toolbox for addressing such questions.

## Methods

**Protein expression.** All protein sequences used in this study are compiled in Supplementary Table 3. All primers (Microsynth) used for site-directed mutagenesis are listed in Supplementary Table 4. The coding sequences of Avi-tagged single-cysteine NCBD and ACTR variants for immobilization were cloned via BamHI/HindIII into a pAT222-pD expression vector (gift of J. Schöppe and A. Plückthun)[54], yielding an expression construct with an N-terminal Avi-tag and a Thrombin-cleavable C-terminal His$_6$-tag. The NCBD-Avi P20A mutant was generated by site-directed mutagenesis of the NCBD-Avi construct. pBirAcm (Avidity) was co-transfected for in vivo biotinylation of Lys12 in the Avi-tag, and expression was carried out in *E.coli* C41(DE3) (Merck). Cells were grown at 37 °C in TYH medium (for 1 l: 20 g trypton, 10 g yeast extract, 11 g HEPES, 5 g NaCl, 1 g MgSO$_4$, pH 7.3), supplied with 0.5% (w/v) glucose, until they reached an OD600 of 0.8. Then, 50 µM biotin in 10 mM bicine buffer (pH 8.3) and 1 mM IPTG were added to the culture. Expression continued for 3 h at 37 °C, after which cells were harvested by centrifugation. The harvested cells were lysed by sonication and the His$_6$-tagged protein enriched via IMAC on Ni-IDA resin (ABT). The His$_6$-tag was then cleaved off with Thrombin (Serva Electrophoresis) and separated from the protein by another round of IMAC. Finally, biotinylated protein was separated from impurities and non-biotinylated protein via reversed-phase HPLC (RP-HPLC) on a C18 column (Reprosil Gold 200, Dr. Maisch) with a H$_2$O/0.1% TFA—acetonitrile gradient and lyophilized.

Single-cysteine variants of NCBD and ACTR were generated by site-directed mutagenesis and co-expressed[55] from a pET-47b(+) vector. The expression construct contained an N-terminal His$_6$-tag cleavable with HRV 3C protease. Transformed *E.coli* C41(DE3) cells were grown at 37 °C in TYH medium supplied

with 0.5% glucose until they reached an OD600 of 0.8, after which expression was induced with 1 mM IPTG for 1 h at 37 °C. Cell harvest, lysis, and protein enrichment via IMAC were carried out as described above, followed by enzymatic cleavage of the His$_6$-tag with HRV 3C protease and separation of the tag from the proteins via another round of IMAC. Finally, ACTR and NCBD were separated with RP-HPLC as described above.

NCBD proline variants were generated by site-directed mutagenesis, co-expressed with ACTR analogously to wild-type NCBD and purified according to the procedures described above.

**Protein labeling.** Lyophilized Avi-tagged protein (NCBD-Avi and ACTR-Avi) was dissolved under nitrogen atmosphere to a concentration of 200 µM in 100 mM potassium phosphate buffer, pH 7.0. The protein was then labeled at room temperature with a 0.8:1 molar ratio of Cy3B maleimide dye (GE Healthcare) to protein. Labeled protein was separated from unlabeled protein with RP-HPLC on a Sunfire C18 column (Waters) as described above. The correct mass of labeled protein was confirmed by electrospray ionization mass spectrometry (ESI-MS).

Analogously, the single-cysteine variants of NCBD and ACTR were labeled with a 1.5-fold molar excess of CF680R maleimide dye (Biotium). The free dye was separated from the labeled protein by RP-HPLC on a C18 column (Reprosil Gold 200) as described above and the protein was lyophilized. The correct mass of labeled protein was confirmed by ESI-MS.

Labeling of the NCBD proline variants was carried out as outlined above, but with a 0.7:1 molar ratio of CF680R dye to protein. The constructs contain two cysteine residues, but as Cys3 is more reactive with maleimides due to its lower pK$_a$, site-specific labeling can be achieved. The single-labeled fraction was separated from unlabeled and double-labeled protein by RP-HPLC (first on a Reprosil Gold 200 column, followed by a Sunfire C18 column). A tryptic digest in combination with ESI-MS confirmed the correct mass and labeling position.

**Surface immobilization.** Adhesive silicone hybridization chambers (Secure Seal Hybridization Chambers, SA8R-2.5, Grace Bio-Labs) were cut in half and bound to PEGylated, biotinylated quartz coverslips (Bio_01, MicroSurfaces, Inc.) to form 150-µl reaction chambers. For surface immobilization experiments, 10 nM NCBD-Avi was pre-coupled to 1 µM Avidin D (Vector Labs) in NaP buffer (50 mM sodium phosphate pH 7.0, 0.01% Tween 20), and immobilized for 10 min in a reaction chamber at a concentration of 20 pM NCBD-Avi/2 nM Avidin D. ACTR-Avi was immobilized by incubating 3 µM Avidin D (Vector Labs) in NaP buffer for 5 min in a reaction chamber, followed by three washing steps with NaP buffer to remove unbound Avidin. Afterwards, the surface was treated with 10 pM ACTR-Avi. With these protocols, a surface coverage of 0.1–0.3 molecules/µm$^2$ was achieved.

**Single-molecule instrumentation and experiments.** All single-molecule experiments were conducted on a custom-built confocal instrument[56] equipped with a 532 nm cw laser (LaserBoxx LBX-532-50-COL-PP, Oxxius) and a 635 nm diode laser (LDH-D-C-635M, PicoQuant). Fluorescence photons were separated from scattered photons with a triple band mirror (zt405/530/630rpc, Chroma) and split onto two channels with a dichroic mirror (T635LPXR, Chroma) according to their wavelength. Donor photons were filtered with an ET585/65 m bandpass filter (Chroma) before detection on a τ-SPAD avalanche photodiode (PicoQuant). Acceptor photons were filtered with a LP647RU long-pass filter (Chroma) and detected with a SPCM-AQRH-14 single-photon avalanche diode (Perkin Elmer). The objective (×UPlanApo 60/1.20-W, Olympus) was mounted onto a piezo stage (P-733.2 and PIFOC, Physik Instrumente GmbH) for scanning.

Binding experiments were conducted at 22 °C under argon atmosphere in NaP buffer supplied with ACTR/NCBD-CF680R. 1% (w/v) glucose, 20 µg/ml glucose oxidase (Sigma) and 20 U/ml catalase (Sigma) were included as oxygen scavenging system, as well as 1 mM methyl viologen and 1 mM ascorbic acid as triplet quenchers[57]. Human Cyclophilin A with an N-terminal FLAG-tag was prepared according to published procedures[58] and included in the measurements shown in Supplementary Fig. 5. Single immobilized NCBD/ACTR molecules were localized by scanning a 16 × 16 µm$^2$ area (130 nm/pixel) with a 532-nm cw laser at a power of 4 µW (measured at the back aperture of the objective). The objective collar and z-positioning of the focus were optimized for molecular brightness. For the acquisition of binding-unbinding time traces, the laser power was reduced tenfold. Time traces for 50–150 immobilized molecules were recorded until the Cy3B dye photobleached. Data were acquired with the SymphoTime software (PicoQuant).

**Analysis of single-molecule time traces.** Single-molecule time traces were inspected to ensure that no substantial brightness variations were occurring (e.g., caused by a drift of the molecule's position, long-lived dark states, or background fluctuations). Suitable traces were analyzed until photobleaching of the donor dye. Single-step photobleaching indicated that only one molecule was present in the confocal volume.

On first inspection, the time traces reveal jumps between two states of low and high FRET efficiency. Hence, in a first analysis step, we approximate the system

with two states, unbound and bound, whose exchange is described with the rate matrix

$$\mathbf{K}_{2\text{state}} = \begin{pmatrix} -k'_{\text{on}} & k_{\text{off}} \\ k'_{\text{on}} & -k_{\text{off}} \end{pmatrix} \qquad (1)$$

$k'_{\text{on}}$ and $k_{\text{off}}$ are the transition rate coefficients between the bound and the unbound state of the immobilized protein. Note that for analyzing the data, we use the pseudo-first-order rate coefficients, $k'_{\text{on}} = k_{\text{on}} \cdot c_{\text{ligand}}$, the product of the second-order rate coefficient, $k_{\text{on}}$, and the concentration of acceptor-labeled binding partner, $c_{\text{ligand}}$. We determined the donor and acceptor photon rates associated with each state by applying the MLH method on binned photon time traces rather than individual photons to achieve sufficient numerical efficiency for a global analysis of all measurements[29,59]. The likelihood, $L_m$, of a time trace $m$ with bin size $\Delta$, number of bins $T_m$, and $(N_{\text{D},t}, N_{\text{A},t})$ donor and acceptor photons detected in time bin $t$ is calculated from

$$L_m = \mathbf{1}^T \left[ \prod_{t=1}^{T_m} \mathbf{F}_t e^{\Delta \mathbf{K}} \right] \mathbf{p}_{\text{eq}} \qquad (2)$$

Here, the population vector $\mathbf{p}_{\text{eq}}$ describes the equilibrium distribution of states for which $\mathbf{K}\mathbf{p}_{\text{eq}} = 0$. $\mathbf{F}_t$ is a diagonal matrix with elements $(\mathbf{F}_t)_{ii} = \frac{(n_{\text{D},i}\Delta)^{N_{\text{D},t}}}{N_{\text{D},t}!} e^{-n_{\text{D},i}\Delta} \times \frac{(n_{\text{A},i}\Delta)^{N_{\text{A},t}}}{N_{\text{A},t}!} e^{-n_{\text{A},i}\Delta}$, assuming Poisson statistics for the number of photons per bin. $n_{\text{D},i}$ and $n_{\text{A},i}$ are the mean photon detection rates of the $i$th state in the donor and acceptor channel, respectively. We found the photon detection rates and transition rate coefficients for each individual time trace (binned at 20 ms) by maximizing $\ln(L_m)$.

In the next step, we used the photon rates and transition rate coefficients to identify the most likely state trajectories using the Viterbi algorithm[60,61]. For this purpose, the photon time traces were binned at 1 ms to avoid averaging over fast events. If the rate matrix $\mathbf{K}_{2\text{state}}$ is used as input, occasional short blinking events are misrecognized as binding events. Therefore, we modify $\mathbf{K}_{2\text{state}}$ to include a dark state accounting for blinking in the low-FRET-efficiency unbound state, which is populated and depopulated with rate coefficients $k_{+\text{b}}$ and $k_{-\text{b}}$:

$$\mathbf{K}_{2\text{state,blink}} = \begin{pmatrix} -(k'_{\text{on}} + k_{+\text{b}}) & k_{\text{off}} & k_{-\text{b}} \\ k'_{\text{on}} & -k_{\text{off}} & 0 \\ k_{+\text{b}} & 0 & -k_{-\text{b}} \end{pmatrix} \qquad (3)$$

Blinking also occurs in the high-FRET-efficiency bound state but does not need to be included in the model since it is not misrecognized as a transition. We note that blinking events can be well distinguished from binding and dissociation transitions, since they have a lower photon rate both in the donor and the acceptor channel (see Supplementary Fig. 2f, l). MLH analysis yields rate coefficients of $k_{+\text{b}} = 1.7\,\text{s}^{-1}$ and $k_{-\text{b}} = 38\,\text{s}^{-1}$ for immobilized NCBD and $k_{+\text{b}} = 5\,\text{s}^{-1}$ and $k_{-\text{b}} = 200\,\text{s}^{-1}$ for immobilized ACTR. From the identified state trajectory, dwell times in the bound ($\tau_{\text{on}}$) and unbound state ($\tau_{\text{off}}$) are obtained to construct 2D-plots of $\langle\tau_{\text{on}}\rangle$ vs. $\langle\tau_{\text{off}}\rangle$ (Figs. 1d, 2d, 3g) and dwell-time histograms (Figs. 2c, 3b, f). Even though the underlying kinetics are clearly more complex than two-state, the Viterbi algorithm still reliably identifies over 92% of all transitions (as shown for simulated data in Supplementary Fig. 6).

Depending on the complexity of the underlying kinetics, the strategy to fit the individual transition rate coefficients is adjusted. NCBD constructs with the P20A mutation (Fig. 3) exhibit single-exponential binding and dissociation kinetics, so the rate matrix $\mathbf{K}_{2\text{state,blink}}$ was used to obtain rate coefficients. We maximize $\sum_m \ln(L_m)$, the sum over the likelihood of all time traces (binned at 1 ms), with respect to $k'_{\text{on}}$ and $k_{\text{off}}$. The mean photon rates of the individual time traces were fixed to the values found in the previous step. NCBD constructs without the P20A mutation exchange slowly between a high-affinity *trans*-proline state and a low-affinity *cis*-proline state. Surface-immobilized ACTR is hence observed in three states: unbound, bound to NCBD1, or bound to NCBD2 (see Fig. 2e). Including donor blinking of the unbound state, we obtain the rate matrix:

$$\mathbf{K}_{3\text{state,blink}} = \begin{pmatrix} -(k'_{\text{on,1}} + k'_{\text{on,2}} + k_{+\text{b}}) & k_{\text{off,1}} & k_{\text{off,2}} & k_{-\text{b}} \\ k'_{\text{on,1}} & -k_{\text{off,1}} & 0 & 0 \\ k'_{\text{on,2}} & 0 & -k_{\text{off,2}} & 0 \\ k_{+\text{b}} & 0 & 0 & -k_{-\text{b}} \end{pmatrix} \qquad (4)$$

The MLH analysis was carried out as described above, with the two bound states having identical photon rates, as suggested by the experimental data.

Finally, if wild-type NCBD is immobilized instead of ACTR (Fig. 1), we observe four states (see Fig. 1e): unbound NCBD1, unbound NCBD2, NCBD1 bound to ACTR, and NCBD2 bound to ACTR. Including donor blinking of

the unbound states, we obtain:

$$\mathbf{K}_{4\text{state,blink}} =$$
$$\begin{pmatrix} -(k_{12} + k'_{\text{on,1}} + k_{+\text{b}}) & k_{\text{off,1}} & k_{-\text{b}} & k_{21} & 0 & 0 \\ k'_{\text{on,1}} & -k_{\text{off,1}} & 0 & 0 & 0 & 0 \\ k_{+\text{b}} & 0 & -(k_{-\text{b}} + k_{12}) & 0 & 0 & k_{21} \\ k_{12} & 0 & 0 & -(k_{21} + k'_{\text{on,2}} + k_{+\text{b}}) & k_{\text{off,2}} & k_{-\text{b}} \\ 0 & 0 & 0 & k'_{\text{on,2}} & -k_{\text{off,2}} & 0 \\ 0 & 0 & k_{12} & k_{+\text{b}} & 0 & -(k_{-\text{b}} + k_{21}) \end{pmatrix}$$
$$(5)$$

Switching events between *cis* and *trans* are rare, do not entail a significant change in transfer efficiency of the bound state, and can only be recognized by the persistent change of the binding kinetics of NCBD. A good estimate for the rate coefficients associated with binding ($k'_{\text{on,1}}$, $k_{\text{off,1}}$, $k'_{\text{on,2}}$, $k_{\text{off,2}}$) is thus needed for a reliable fit of the much lower isomerization rate coefficients, $k_{12}$ and $k_{21}$. We obtain these estimates by an iterative procedure, where time traces are initially split into *cis* and *trans* segments for separate analysis of the binding kinetics. In the first step, we estimate the blinking rate coefficients, $k_{+\text{b}}$ and $k_{-\text{b}}$, by maximizing the likelihood of the kinetic model based on the rate matrix $\mathbf{K}_{2\text{state,blink}}$, and fix them in all further steps. Second, we fix $k'_{\text{on,1}}$, $k_{\text{off,1}}$, $k'_{\text{on,2}}$ and $k_{\text{off,2}}$ to the values estimated from the $\langle\tau_{\text{on}}\rangle$ vs. $\langle\tau_{\text{off}}\rangle$ plot (Fig. 1d) and estimate the *cis/trans*-isomerization rate coefficients, $k_{12}$ and $k_{21}$, individually. Next, we identify the most likely state trajectory with the Viterbi algorithm, fixing the rate coefficients to the previously estimated values. Now, the time traces are split according to their *cis* or *trans* state identified by the Viterbi algorithm. The *cis* and *trans* segments are analyzed with the rate matrix $\mathbf{K}_{2\text{state,blink}}$ to obtain $k'_{\text{on,1}}$ and $k_{\text{off,1}}$ (from the *trans* segments) as well as $k'_{\text{on,2}}$ and $k_{\text{off,2}}$ (from the *cis* segments). Finally, this set of refined rate coefficients is used in a second iteration to fit the isomerization rate coefficients, $k_{12}$ and $k_{21}$, with the model $\mathbf{K}_{4\text{state,blink}}$ in a global analysis of all time traces.

**Fluorescence correlation spectroscopy.** Before and after recording time traces, the concentration of CF680R-labeled ligand in solution was estimated by measuring FCS curves with the 635-nm diode laser at a power of 10 μW (measured at the back aperture of the objective). The laser was focused into the solution, 20 μm above the surface. Fluorescence photons were separated according to their polarization and detected on two channels. The fluorescence signal of both channels was cross-correlated. The amplitude of the cross-correlation of the acceptor signal, $G_{\text{AA}}(\tau)$, was used to estimate the concentration of the CF680R-labeled binding partner in the surface experiments. The amplitude of $G_{\text{AA}}(\tau)$ depends on the average number of molecules $\langle N \rangle$ in the confocal volume, which is proportional to the concentration of the fluorescent species in the solution[62]. Occasional high-intensity fluorescence bursts caused by protein aggregates were removed from the raw acceptor signal. Then, the mean number of molecules in the confocal volume was obtained by fitting the FCS curve with a model including terms for translational diffusion and triplet dynamics:

$$G_{\text{AA}}(\tau) = 1 + \frac{1}{\langle N \rangle}\left(1 + \frac{\tau}{\tau_{\text{D}}}\right)^{-1}\left(1 + s^2\frac{\tau}{\tau_{\text{D}}}\right)^{-1/2}\left(1 + c_{\text{T}}\exp\left[-\frac{\tau}{\tau_{\text{T}}}\right]\right) \qquad (6)$$

The parameter $s$ (describing the aspect ratio of the confocal volume) was set to 0.27; the triplet time $\tau_{\text{T}}$ and the triplet amplitude $c_{\text{T}}$ were averaged over all fits and used as constants in a second iteration of the fit. The remaining free parameters were the diffusion time, $\tau_{\text{D}}$, and the mean number of molecules in the confocal volume, $\langle N \rangle$. A calibration curve[63] was recorded to convert $\langle N \rangle$ to the actual concentration of acceptor-labeled species. Two independent FCS curves (before and after the recording of time traces) were measured and averaged to obtain concentrations. The variation in $\langle N \rangle$ between those two measurements was <5%.

**Stopped-flow experiments.** Stopped-flow experiments were conducted either with biotinylated NCBD-Avi-Cy3B and ACTR-CF680R or with biotinylated ACTR-Avi-Cy3B and NCBD-CF680R in NaP buffer at 20 °C on a PiStar-180 spectrometer (Applied Photophysics). Cy3B-labeled protein was excited with the 546-nm line of an Hg/Xe lamp and the loss of donor fluorescence after binding was monitored. Fluorescence emission was filtered with a 550-nm long-pass filter (the concomitant increase in acceptor fluorescence could not be detected, as the photomultiplier tube is not sensitive to emission wavelengths >650 nm). Concentrations of labeled protein were monitored via absorbance at 680 nm before and after the stopped-flow measurement to correct for adhesion of protein to the glass syringe or tubing. In one experiment, 105 nM of donor-labeled NCBD were mixed with 130, 260, 530, and 1540 nM of acceptor-labeled ACTR. In the other experiment, 390 nM of donor-labeled ACTR were mixed with 250, 570, and 1600 nM of acceptor-labeled NCBD. The mixing ratio was always kept at 1:2.5 (donor:acceptor), with final concentrations of 30 nM NCBD-Avi-Cy3B or 110 nM ACTR-Avi-Cy3B as the non-varied species after mixing.

**MD simulations.** MD simulations were performed using GROMACS[64] version 2016.4 (http://www.gromacs.org/). In a first step, the NCBD-ACTR complex was

equilibrated. The initial structure of the bound NCBD-ACTR complex from the NMR ensemble was used (PDB accession code: 1KBH)[21]. The molecular topology of the complex was created according to the parameters from the AMBER12 force field[65], and the complex was placed in a dodecahedral box solvated with water molecules parameterized according to the TIP4PD water model[66]. This combination of force field and water model has proven effective for sampling the dynamics of IDPs[66]. $Na^+$ and $Cl^-$ ions were added to the solution to neutralize the net charge of the complex and reach an ionic strength of 0.15 M. The system was then energy-minimized using a steep descent algorithm with an initial step size of 0.01 nm for 5000 steps or when the maximum force converged below 10 kJ $mol^{-1}$ $nm^{-1}$. The solvent was then equilibrated in two runs, each 500 ps long. A first equilibration run was performed in the nVT ensemble, in which the temperature was coupled every 0.1 ps at a value of 300 K using the V-rescale thermostat. Random velocities drawn from a Boltzmann distribution obtained at 300 K were assigned to each particle of the system. The second equilibration run was in the nPT ensemble, in which temperature and pressure, coupled isotropically, were kept constant at values of 300 K and 1 bar using the V-rescale thermostat and the Parrinello-Rahman barostat, respectively. Temperature was coupled every 0.1 ps, whereas pressure was coupled every 2.0 ps. In both the nVT and nPT runs, protein atoms were subject to a position-restraining potential of 1000 kJ $mol^{-1}$ $nm^2$. Subsequent to solvent equilibration, a 500-ns MD production run was performed in order to equilibrate the protein complex, using a time step of 2 fs. Interactions were calculated by building pair-lists using the Verlet scheme. Van-der-Waals (vdW) and Coulomb interactions were computed using a cutoff of 1.4 nm. The use of interaction cutoffs longer than 1.0 nm has been shown to improve the sampling of the conformational dynamics of IDPs[67]. Beyond the cutoff, electrostatics were treated using the Particle Mesh Ewald (PME) algorithm using a grid spacing of 0.16 and a cubic B-spline interpolation level of 4.

The last conformer obtained from the equilibration run was used as starting point for well-tempered metadynamics simulations[68], which were performed to generate initial conformations of NCBD with Pro20 either in *trans* or in *cis*. Thus, the ω angle formed by the atoms $^{S19}C_\alpha$-$^{S19}C$-$^{P20}N$-$^{P20}C_\alpha$ was used as the reaction coordinate, on which a history-dependent bias represented by Gaussian potential functions was applied. The simulations were carried out using GROMACS version 2016.4 patched with the Plumed plugin version 2.3[69] by adding Gaussian potentials of height equal to 0.2 kJ $mol^{-1}$ and width of 0.2 radians at a temperature of 300 K, using as bias factor a value of 6 and a Gaussian deposit time of 1 ps.

To sample the conformational dynamics of the NCBD-ACTR complex at equilibrium in either the *trans* or *cis* state, two conformations with Pro20 either in *trans* or in *cis* (RMSD difference 0.25 nm) were taken from the well-tempered metadynamics run and used as starting points for temperature replica exchange MD. The systems were prepared as in the run performed to initially equilibrate the system (as described above), and the solvent was equilibrated again in the nVT and nPT ensembles at 14 different temperatures between 298.15 K and 323.11 K. The equilibrated replicas were run for 300 ns each using a time step of 2 fs, and replicas were allowed to swap every 500 steps (every ps). The number of replicas and the respective set of temperatures were chosen using a temperature predictor[70] with the aim to achieve an exchange rate of at least 20% between replicas, with the exchange probability calculated for each replica after the simulations being between 21 and 22%. The analysis of the collected trajectory was then carried out considering the replica at 298.15 K using either GROMACS or VMD (https://www.-s.ks.uiuc.edu/Research/vmd/) tools, as well as in-house analysis scripts. Contact analysis was performed using the program CONAN[71]. Hydrogen bonds were identified by choosing a distance between polar hydrogens and hydrogen acceptors lower than or equal to 0.3 nm and a donor–hydrogen–acceptor angle of less than 20°.

**Data availability**. Data supporting the findings of this manuscript are available from the corresponding author upon reasonable request. A custom module for Mathematica (Wolfram Research) used for the analysis of single-molecule fluorescence data is available upon request.

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

## Acknowledgements

We thank Jane Dyson, Peter Wright, Kaare Teilum, and Magnus Kjærgaard for the discussion of peak splitting due to peptidyl-prolyl *cis/trans* isomerization in their NMR data; Attila Szabo for the discussion of single-molecule kinetics; Damien Morger, Maximilian Mittelvieshaus, and Markus Grütter for the gift of Cyclophilin A; Jendrik Schöppe and Andreas Plückthun for the gift of a pAT222-pD expression vector; Tina Schelbert and Karin Buholzer for help with protein expression; and the Functional Genomics Center Zurich for expert mass spectrometry analysis. This work was supported by the Swiss National Science Foundation (grant numbers 200021_169741, 310030B_173333, CRSII5_170976). Molecular dynamics simulations were carried out on Piz Daint at the CSCS Swiss National Supercomputing Centre.

## Author contributions

F.Z. and B.S. designed the research. F.Z. prepared the protein samples, conducted experiments, and analyzed and interpreted the data. D.M. conducted and analyzed the MD simulations. D.N. developed instrumentation and data analysis tools, and assisted with data analysis. B.S. supervised the work. F.Z. and B.S. wrote the paper with the help of the other authors.

## Additional information

**Competing interests:** The authors declare no competing interests.

