## [Peer Review File · Nature Communications]

Reviewers' comments:

Reviewer #1 (Remarks to the Author):

There is currently a great interest in the role of protein-protein interactions in cellular regulation, and in particular how conformational heterogeneity influences these interactions. Zosel et al address this question of distinct functional states in protein-protein interaction domains. More specifically, they demonstrate that a cis-trans isomerization results in different binding properties for the two resulting conformations of an interaction domain called NCBD. This interaction domain has previously been shown to interact with several different binding partners. Therefore, this "Pro switch", which is evolutionarily conserved, could be a way for NCBD to better accommodate its different protein ligands.

The authors use state of the art single molecule techniques in combination with mutagenesis and molecular dynamics simulation to show that a specific Pro residue is responsible for switching between a higher and a lower affinity conformation (with regard to one of the ligands of NCBD called ACTR). Since Pro residues are enriched in intrinsically disordered proteins cis-trans isomerization could be a common mechanism among IDPs to modulate affinity toward different binding partners and - as the authors write - understanding the mechanism is important for understanding function. To understand mechanism kinetic approaches such as the one described here are essential.

The kinetic experiments and the interpretation of data are sound and the conclusions are supported by data. It is often argued that extrinsic labels will affect the results and this is clearly the case here. However, by using different labeling and mutagenesis it is clear that the conclusions regarding the Pro switch are valid. I have some minor points to consider:

1. Introduction, end of 2nd paragraph: "All of these [intramolecular] dynamics are faster than the typical kinetics of binding at cellular protein concentrations." Is there a reference for this? While the statement is probably true I think we know very little about local concentrations of ligands in the cell.

2. Discussion, p12, line 5; add "coefficient" after dissociation rate and association rate.

3. Discussion, last sentence of p.12; replace nonexponential kinetics with multiexponential kinetics.

4. Discussion, first sentence of p.13; I suggest rewriting the sentence to something like: ...that enables the formation of interactions by combining charge-promoted [rather than driven] association with dissociation...

(What is "transient" and what is "high affinity"? And however we define the terms they will to some extent contradict each other.)

5. All experimental conditions must be clearly defined. For example, I could not find buffer conditions and temperature for the single molecule experiments and only buffer (no concentration) and temperature for the stopped-flow experiments.

6. Stopped-flow experiments, methods, a mixing ratio of 2.5:1 is given, but I assume the final concentration of non-varied species was 30 and 110 microM, respectively, at each concentration of the varied species?

7. Supplementary Fig. 4e; There is a clear trend in the residuals when NCBD is in excess of ACTR but not when ACTR is in excess. Is this reflecting the two subpopulations of NCBD? That is, if you fit to a double exponential, will you get two observed rate constants consistent with the single

molecule experiments?

Reviewer #2 (Remarks to the Author):

This is a very exciting paper describing quantitatively the kinetics of association of two intrinsically disordered proteins (IDP) that form a structure upon binding. The authors find that proline cis/trans isomerization has a pronounced effect on the coupled folding and binding process and can explain the observed kinetic heterogeneity. The phenomena described in this paper may as well be relevant for many IDPs given that there is high content of Pro residues in IDPs. The two proteins studied are ACTR, which lacks structure as a monomer, and NCBD, which is a marginally stable molten globule.

Experimentally, this system is probed using single molecule FRET spectroscopy. FRET experiments find that NCBD occupies two distinct subpopulations at equilibrium that differ by the conformation of a proline residue. The manuscript reports extensive measurements and controls to convincingly identify Pro 20 as the switching residue responsible for the two subpopulations. The single molecule data analysis is exhaustive and convincing.

The experimental results are complemented by MD simulations of ACTR and NCBD (in the cis and trans conformations for Pro20). The cis/trans switching is studied by well-tempered metadynamics. The cis/trans equilibrium is sampled by replica exchange MD (REMD) simulations. The simulations show an increase in the fluctuations of NCBD when Pro 20 is in the trans state that are driven by loss of packing in the hydrophobic core. The calculations suggest that the fast dissociation and reduce affinity are consistent with the long-range structural changes induced by the change of isomerization by Pro20. These results are summarized in Figure 4 and Figure S7.

It is not clear why the authors did REMD and WT-metadynamics, since the results are not presented in manuscript. The statistical analysis for the simulations is described for Pro20 in the cis or trans. However, there is no description of the equilibrium or conversion between cis/trans conformations. This may be a good thing since it is not clear that the force field used can describe the energetics of the cis/trans isomerization. Work by Neale et al. (JCTC 12, 1989 (2016) show that for a simple peptide undergoing cis/trans isomerization, REMD cannot reach equilibrium populations at low Ts (~300K). The work by Neale et al. did not include Amber12 force field in their analysis. Is it known if Amber12 describe cis/trans isomerization correctly? If not, the authors should show that their results are equilibrated (i.e., the distributions are representative of the energetics and have no memory effects) and provide an uncertainty analysis for their calculations.

Overall, the manuscript describes outstanding and very important results describing the role of Pro isomerization on the dissociating kinetics of IDP complexes. This effect may be important for many IDPs. The experimental measurements and analysis is outstanding and of very high quality. The simulation results could be improved.

Reviewer #3 (Remarks to the Author):

The interesting work by Zosel et al. demonstrates the effects of cis-trans proline isomerization on the binding reaction of two intrinsically disordered proteins, NCBD and ACTR. Proline isomerization effects on protein folding have been studied for decades, however the effects on IDP binding are less well understood. This study uses a series of experiments using single molecule FRET and mutational analysis to discover and validate a significant effect of isomerization of a single proline residue in NCBD on this binding equilibrium. Simulations then indicate that the proline isomers result in different dynamics of the complexes. The work is carefully done with many controls and

alternative methods used to study the problem. The result, while in a single system, has interesting implications for binding of many other IDPs. The work also provides another nice example of a system where the multiple states are inferred not from very small differences in observed FRET efficiencies, but from "memory effects" in corresponding reactions from these states, showing a strength of the single molecule approach. Overall, the new insights obtained into a problem of broad interest make this work of interest to the broad readership of Nature Communications. I only have a couple of few minor concerns noted below that the authors may consider to further improve the paper.

It could be considered that a weakness of the study is that the surface attachment strategy seems to perturb the system. Indeed, as the authors note, the association rate coefficients are a few fold different depending on which protein is immobilized. Based on their comparisons with solution ensemble experiments, the origin of this difference seems to be in the biotin incorporation of the negatively charged Avi tag for biotinylation which alters the electrostatically driven association. While there are alternative strategies to get around this issue, I believe that this issue does not in any way take away from the overall results and conclusions of this carefully performed work. The variety of experiments including the ones with the proline substitutions and isomerase strongly validate the authors' conclusions. The authors may want to add a note to this effect in the paper.

It makes sense that binding in the trans Pro20 conformation locks the pro into this isomer for the duration of the complex. However, it would be slightly surprising that cis-to-trans Pro20 isomerization is not occurring given the dynamic nature of the complex. However, as the authors state, this may be because of the reaction is too slow to observe with the performed experiments.

It would be useful to include data showing that addition of Cyclophilin A in the P20 mutants does not result in changes in binding rates.

Response to the reviewers' comments:

We were of course very pleased by the favorable assessments of the reviewers, and we thank them for their insightful comments, which were very helpful for improving the manuscript further. Our response to the reviewers' comments and a detailed description of the changes we made in the manuscript are listed below. All changes are marked in yellow in the manuscript.

Reviewer #1:

There is currently a great interest in the role of protein-protein interactions in cellular regulation, and in particular how conformational heterogeneity influences these interactions. Zosel et al address this question of distinct functional states in protein-protein interaction domains. More specifically, they demonstrate that a cis-trans isomerization results in different binding properties for the two resulting conformations of an interaction domain called NCBD. This interaction domain has previously been shown to interact with several different binding partners. Therefore, this "Pro switch", which is evolutionary conserved, could be a way for NCBD to better accommodate its different protein ligands.

The authors use state of the art single molecule techniques in combination with mutagenesis and molecular dynamics simulation to show that a specific Pro residue is responsible for switching between a higher and a lower affinity conformation (with regard to one of the ligands of NCBD called ACTR). Since Pro residues are enriched in intrinsically disordered proteins cis-trans isomerization could be a common mechanism among IDPs to modulate affinity toward different binding partners and - as the authors write - understanding the mechanism is important for understanding function. To understand mechanism kinetic approaches such as the one described here are essential.

The kinetic experiments and the interpretation of data are sound and the conclusions are supported by data. It is often argued that extrinsic labels will affect the results and this is clearly the case here. However, by using different labeling and mutagenesis it is clear that the conclusions regarding the Pro switch are valid. I have some minor points to consider:

We thank the reviewer for the thorough reading of our manuscript and the helpful points raised. We have incorporated the suggested changes as outlined below.

1. Introduction, end of 2nd paragraph: "All of these [intramolecular] dynamics are faster than the typical kinetics of binding at cellular protein concentrations." Is there a reference for this? While the statement is probably true I think we know very little about local concentrations of ligands in the cell.

Our thinking followed a back-of-the-envelope estimate of the following type: Let us assume a typical total protein concentration of $\sim 10^6$ protein molecules per μm^3 for a eukaryotic cell (see, e.g., Milo 2013, Bioessays 35: 1050–1055). Given that this concentration is expected to be dominated by the ~ 1000 most abundant proteins (Milo 2013, Bioessays 35: 1050–1055), this results in an average concentration of $\sim 1 \mu\text{M}$ for each specific protein. With a dissociation constant of $1 \mu\text{M}$ for a reasonably affine interaction and even a fast association rate coefficient of $10^6 \text{ M}^{-1}\text{s}^{-1}$ (see, e.g.,

Shammas et al., 2016, JBC 291, 6689–6695), both the effective on- and off-rates are in the range of $\sim 1 \text{ s}^{-1}$. Clearly, this value may vary by orders of magnitude from case to case, but it seems reasonable to assume that conformational dynamics will typically occur on faster timescales.

We think that including such a calculation explicitly in the text would be a digression from the main train of thought, but we now included the two references mentioned above along with the statement. To make sure that the reader does not expect our statement to be a universal truth, we also rephrased it slightly to be more cautious: “All of these dynamics are **expected to be** faster than the typical kinetics of binding at cellular protein concentrations.”

2. Discussion, p12, line 5; add "coefficient" after dissociation rate and association rate.

Thank you for picking up this omission. “coefficient” was added.

3. Discussion, last sentence of p.12; replace nonexponential kinetics with multiexponential kinetics.

“nonexponential” was replaced by “multi-exponential”.

4. Discussion, first sentence of p.13; I suggest rewriting the sentence to something like:
...that enables the formation of interactions by combining charge-promoted [rather than driven] association with dissociation...

We agree with this suggestion and replaced “charge-driven” by “charge-promoted”.

(What is "transient" and what is "high affinity"? And however we define the terms they will to some extent contradict each other.)

We agree with this parenthetic remark, but we assume that the reader will be able to see it in context, i.e., that “transient” refers to timescales that enable regulation on a biologically relevant timescale, and that the nanomolar affinity of ACTR/NCBD is fairly high compared to many other protein-protein interactions.

5. All experimental conditions must be clearly defined. For example, I could not find buffer conditions and temperature for the single molecule experiments and only buffer (no concentration) and temperature for the stopped-flow experiments.

We moved the section describing experimental conditions for the single-molecule experiments from the subheading “Surface immobilization” to “Single-molecule instrumentation and experiments” to make this information more easily accessible and added the temperature at which the experiments were conducted.

6. Stopped-flow experiments, methods, a mixing ratio of 2.5:1 is given, but I assume the final concentration of non-varied species was 30 and 110 microM, respectively, at each concentration of the varied species?

The concentrations of the non-varied species were indeed 30 and 110 nanomolar. We conducted the stopped-flow experiments at these low concentrations because the fluorescence of Cy3B-labeled NCBD and ACTR can be detected with high sensitivity. With respect to remarks 5 and 6, we now describe the stopped-flow experiments in more detail and added the following text in the Methods section:

“In one experiment, 105 nM of donor-labeled NCBD were mixed with 130, 260, 530 and 1540 nM of acceptor-labeled ACTR. In the other experiment, 390 nM of donor-labeled ACTR were mixed with 250, 570 and 1600 nM of acceptor-labeled NCBD. The mixing ratio was always kept at 1:2.5 (donor:acceptor), with final concentrations of 30 nM NCBD-Avi-Cy3B or 110 nM ACTR-Avi-Cy3B as the non-varied species after mixing.”

7. Supplementary Fig. 4e; There is a clear trend in the residuals when NCBD is in excess of ACTR but not when ACTR is in excess. Is this reflecting the two subpopulations of NCBD? That is, if you fit to a double exponential, will you get two observed rate constants consistent with the single molecule experiments?

Indeed, the data is better described with a double exponential (as mentioned in the caption), and the trend in the residuals is likely to reflect the binding of the subpopulations of NCBD. A free fit with a double-exponential decay, however, fails to reproduce the ratio of amplitudes and decay rates expected from the single-molecule experiments, probably at least in part because a significant fraction of the fluorescence decay already occurs in the dead time of the instrument. Because of these associated uncertainties, we chose to fit the binding curves with a single-exponential decay.

Reviewer #2

This is a very exciting paper describing quantitatively the kinetics of association of two intrinsically disordered proteins (IDP) that form a structure upon binding. The authors find that proline cis/trans isomerization has a pronounced effect on the coupled folding and binding process and can explain the observed kinetic heterogeneity. The phenomena described in this paper may as well be relevant for many IDPs given that there is high content of Pro residues in IDPs. The two proteins studied are ACTR, which lacks structure as a monomer, and NCBD, which is a marginally stable molten globule.

Experimentally, this system is probed using single molecule FRET spectroscopy. FRET experiments find that NCBD occupies two distinct subpopulations at equilibrium that differ by the conformation of a proline residue. The manuscript reports extensive measurements and controls to convincingly identify Pro 20 as the switching residue responsible for the two subpopulations. The single molecule data analysis is exhaustive and convincing.

The experimental results are complemented by MD simulations of ACTR and NCBD (in the cis and trans conformations for Pro20). The cis/trans switching is studied by well-tempered metadynamics. The cis/trans equilibrium is sampled by replica exchange MD (REMD) simulations. The simulations show an increase in the fluctuations of NCBD when Pro 20 is in the trans state that are driven by loss of packing in the hydrophobic core. The calculations suggest that the fast dissociation and reduce affinity are consistent with the long-range structural changes induced by the change of isomerization

by Pro20. These results are summarized in Figure 4 and Figure S7.

It is not clear why the authors did REMD and WT-metadynamics, since the results are not presented in manuscript. The statistical analysis for the simulations is described for Pro20 in the *cis* or *trans*. However, there is no description of the equilibrium or conversion between *cis/trans* conformations. This may be a good thing since it is not clear that the force field used can describe the energetics of the *cis/trans* isomerization. Work by Neale *et al.* (JCTC 12, 1989 (2016) show that for a simple peptide undergoing *cis/trans* isomerization, REMD cannot reach equilibrium populations at low Ts (~300K). The work by Neale *et al.* did not include Amber12 force field in their analysis. Is it known if Amber12 describe *cis/trans* isomerization correctly? If not, the authors should show that their results are equilibrated (*i.e.*, the distributions are representative of the energetics and have no memory effects) and provide an uncertainty analysis for their calculations.

Overall, the manuscript describes outstanding and very important results describing the role of Pro isomerization on the dissociating kinetics of IDP complexes. This effect may be important for many IDPs. The experimental measurements and analysis is outstanding and of very high quality. The simulation results could be improved.

We thank the reviewer for the insightful suggestions for improving our MD results. Accordingly, we modified the main text to better highlight the reasons for which we chose a combined approach featuring well-tempered metadynamics and replica-exchange molecular dynamics. In this context, we also cite the work of Neale *et al.* to stress that no transitions between *cis* and *trans* are expected in the range of temperatures used:

“We thus modeled the *cis* state by perturbing the ω -angle by well-tempered metadynamics (see Methods for details), which shows an energy barrier between the two states of ~ 80 kJ mol⁻¹, in line with experimental results³⁴ and previous calculations using the AMBER2003 force field³⁵. We then simulated the *cis* and *trans* conformations separately by replica exchange molecular dynamics to investigate the stability of the complex when P20 is in either the *trans* or the *cis* state. Within the simulated time and at the chosen temperatures, the simulations do not sample prolyl isomerization, as expected [ref. to Neale *et al.*] (Supplementary Fig. 7).”

In the new Supplementary Figure 7, we now present the analysis of the well-tempered metadynamics simulations to report the free energy surface of P20 *cis/trans* isomerisation. The sampled barrier between the *cis* and *trans* states matches previous experimental and computational results (references included in the main text), indicating convergence. Furthermore, we report the energy difference between the *cis* and *trans* states sampled by metadynamics as a function of time (inset to panel a) to show that the metadynamics run is well equilibrated after 1 μ s. Additionally, to demonstrate that at the temperatures chosen for each replica, the Amber12 simulation is not sampling isomerization events, we show the value of the S19-P20 ω -angle as a function of the simulation time (panel b). This finding is in line with the previous observations by Neale *et al.* mentioned by the referee, who report spontaneous prolyl isomerization only at T > 500 K by Amber99sb*-ILDN, and provides additional information regarding the performance of Amber12 with respect to prolyl isomerisation at the simulated temperatures.

Reviewer #3

The interesting work by Zosel et al. demonstrates the effects of cis-trans proline isomerization on the binding reaction of two intrinsically disordered proteins, NCBD and ACTR. Proline isomerization effects on protein folding have been studied for decades, however the effects on IDP binding are less well understood. This study uses a series of experiments using single molecule FRET and mutational analysis to discover and validate a significant effect of isomerization of a single proline residue in NCBD on this binding equilibrium. Simulations then indicate that the proline isomers result in different dynamics of the complexes. The work is carefully done with many controls and alternative methods used to study the problem. The result, while in a single system, has interesting implications for binding of many other IDPs. The work also provides another nice example of a system where the multiple states are inferred not from very small differences in observed FRET efficiencies, but from “memory effects” in corresponding reactions from these states, showing a strength of the single molecule approach. Overall, the new insights obtained into a problem of broad interest make this work of interest to the broad readership of Nature Communications. I only have a couple of few minor concerns noted below that the authors may consider to further improve the paper.

It could be considered that a weakness of the study is that the surface attachment strategy seems to perturb the system. Indeed, as the authors note, the association rate coefficients are a few fold different depending on which protein is immobilized. Based on their comparisons with solution ensemble experiments, the origin of this difference seems to be in the biotin incorporation of the negatively charged Avi tag for biotinylation which alters the electrostatically driven association. While there are alternative strategies to get around this issue, I believe that this issue does not in any way take away from the overall results and conclusions of this carefully performed work. The variety of experiments including the ones with the proline substitutions and isomerase strongly validate the authors’ conclusions. The authors may want to add a note to this effect in the paper.

We thank the reviewer for appreciating the strengths of our manuscript and the suggestions to improve it further. To stress the aspect regarding the variety of complementary experiments validating the conclusions, we added the following sentence in the Discussion:

“Cis/trans-isomerization as the origin of the kinetic heterogeneity observed in single-molecule experiments is confirmed by proline substitutions and the effect of proline cis/trans-isomerase on the kinetics.”

It makes sense that binding in the trans Pro20 conformation locks the pro into this isomer for the duration of the complex. However, it would be slightly surprising that cis-to-trans Pro20 isomerization is not occurring given the dynamic nature of the complex. However, as the authors state, this may be because of the reaction is too slow to observe with the performed experiments.

As the reviewer points out correctly, the kinetics of cis/trans isomerization within the complex are difficult to quantify directly with our experimental approach. The energy barrier for prolyl

isomerization in NCBD in the bound state is very likely to be higher than in the unbound state owing to the restricted degrees of freedom, resulting in even lower rates. The only chance to observe such events would be under conditions where NCBD is in the bound state almost all the time – conditions under which our experimental approach would fail because we rely on being able to clearly distinguish binding and dissociation transitions in the single-molecule time traces. However, a confirmation of our inference based on the rates we determined comes from the absence of a *cis* Pro isomer in the NMR structure of the complex. We now stress this point by slightly rephrasing the corresponding statement in the text:

“Based on the parameters we obtained here (Fig. 5), the *cis* complex is only populated to 6% at equilibrium at the concentrations of 3 mM NCBD and 6 mM ACTR where the structure was solved. This small *cis* population is consistent with the NMR structure of the NCBD-ACTR complex, where only one population with Pro20 in *trans* was observed²².”

It would be useful to include data showing that addition of Cyclophilin A in the P20 mutants does not result in changes in binding rates.

In Supplementary Figure 5 and Supplementary Table 1, we compare the association and dissociation rate coefficients in the presence and absence of Cyclophilin A. We observe that the only affected rates are k_{12} and k_{21} , associated with *cis/trans* isomerization, whereas the binding rate coefficients remain the same within error. We see no reason to assume that this behavior would be different in the P20 mutants, so we think that additional experiments are dispensable. However, we modified the discussion in the main text to clarify this point:

“Cyclophilin A accelerates the switching between low- and high-affinity subpopulations of NCBD (k_{12} and k_{21}) by almost an order of magnitude, while leaving the association rate coefficients unaffected (Supplementary Fig. 5, Supplementary Table 1).”

Additional changes included to comply with manuscript guidelines

- The abstract was shortened.
- The subheadings in the Results section were shortened.
- Six references were removed, and an additional Supplementary Figure (Supplementary Fig. 9) showing the sequence conservation of NCBD was included for clarity.
- A code availability statement was added.

REVIEWERS' COMMENTS:

Reviewer #1 (Remarks to the Author):

I am happy with the rebuttals and the revision

Reviewer #2 (Remarks to the Author):

The authors have addressed all my comments and concerns. This is an outstanding manuscript. I recommend publication.

Reviewer #3 (Remarks to the Author):

The authors have done a good job addressing my previous comments and suggestions. I believe that the revised manuscript will be of high interest to the readers of Nature Communications.